# Chemical Transformation of α-Pinene derived Organosulfate via Heterogeneous OH Oxidation: Implications for Sources and Environmental Fates of Atmospheric Organosulfates

Rongshuang Xu[1], Sze In Madeleine Ng[1], Wing Sze Chow[2], Yee Ka Wong[3], Yuchen Wang[2], Donger Lai[1], Zhong-Ping Yao[4], Pui-Kin So[5], Jian Zhen Yu[2*], and Man Nin Chan[1,6*]

[1]Earth System Science Programme, Faculty of Science, The Chinese University of Hong Kong, Hong Kong, China

[2]Department of Chemistry, The Hong Kong University of Science and Technology, Hong Kong, China

[3]Division of Environment and Sustainability, The Hong Kong University of Science and Technology, Hong Kong, China

[4]State Key Laboratory of Chemical Biology and Drug Discovery and Department of Ap-

plied Biology and Chemical Technology, The Hong Kong Polytechnic University, Hong Kong, China

[5]The University Research Facility in Life Sciences, The Hong Kong Polytechnic University, Hong Kong, China

[6]The Institute of Environment, Energy, and Sustainability, The Chinese University of Hong

Kong, Hong Kong, China

*Corresponding to: Jian Zhen Yu (jian.yu@ust.hk); Man Nin Chan (mnchan@cuhk.edu.hk)

**Abstract**.

Organosulfur compounds are found to be ubiquitous in atmospheric aerosols — a majority of which are expected to be organosulfates (OSs). Given the atmospheric abundance of OSs, and their potential to form a variety of reaction products upon ageing, it is imperative to study the transformation kinetics and chemistry of OSs to better elucidate their atmospheric fates and impacts. In this work, we investigated the chemical transformation of an α-pinene

derived organosulfate ($C_{10}H_{17}O_5SNa$, αpOS-249) through heterogeneous OH oxidation at a relative humidity of 50 % in an oxidation flow reactor (OFR). The aerosol-phase reaction products were characterized using the high-performance liquid chromatography-electrospray ionization-high resolution mass spectrometry and the ion chromatography. By monitoring the decay rates of αpOS-249, the effective heterogeneous OH reaction rate was

measured to be $(6.72 \pm 0.55) \times 10^{-13}$ cm$^3$ molecule$^{-1}$ s$^{-1}$. This infers an atmospheric lifetime

of about two weeks at an average OH concentration of $1.5 \times 10^6$ molecules cm$^{-3}$. Product analysis shows that OH oxidation of αpOS-249 can yield more oxygenated OSs having a nominal mass-to-charge ratio ($m/z$) at 247 ($C_{10}H_{15}O_5S^-$), 263 ($C_{10}H_{15}O_6S^-$), 265 ($C_{10}H_{17}O_6S^-$), 277 ($C_{10}H_{13}O_7S^-$), 279 ($C_{10}H_{15}O_7S^-$), and 281 ($C_{10}H_{17}O_7S^-$). The formation of fragmentation products, including both small OSs (C < 10) and inorganic sulfates, is found to be insignificant. These observations suggest that functionalization reactions are likely the dominant processes and that multigenerational oxidation possibly leads to formation of products with one or two hydroxyl and carbonyl functional groups adding to αpOS-249. Furthermore, all product ions except $m/z = 277$ have been detected in laboratory generated α-pinene derived secondary organic aerosols as well as in atmospheric aerosols. Our results reveal that OSs freshly formed from the photochemical oxidation of α-pinene could react further to form OSs commonly detected in atmospheric aerosols through heterogeneous OH oxidation. Overall, this study provides more insights into the sources, transformation, and fate of atmospheric OSs.

## 1. Introduction

Sulfur-containing aerosols are of particular significance for human health because of their high abundance and significant impacts on regional air quality and global climate (Bentley et al., 2004; Riva et al., 2015; Stadtler et al., 2018). The environmental and climatic impacts of inorganic sulfate aerosols such as sulfate are well known. Recently, organosulfur compounds originating from terrestrial, marine, and anthropogenic emissions have also been found to be a significant component of atmospheric aerosols (Tolocka et al., 2012; Huang et al., 2015; Shakya et al., 2013, 2015). Organosulfates (OSs) have been found to be the most important class of organosulfur compounds (Brüggemann et al., 2020), and can greatly affect aerosol formation and growth by varying their surface activity, water uptake, and their ability to serve as cloud condensation nuclei (Hansen et al., 2015; Vogel et al., 2016). Various formation pathways and precursors of OSs have been identified, including biogenic and anthropogenic volatile organic compounds (VOCs) such as isoprene, monoterpenes such as α-pinene, β-pinene, and limonene, oxygenated VOCs, and even aromatic compounds (Iinuma et al., 2005, 2007; Surratt et al., 2008; Kristensen et al., 2011; Zhang et al., 2012; Riva et al., 2015). Given their ubiquity and atmospheric significance, it is crucial to understand the fates and subsequently environmental impacts of OSs. However, although the formation mechanisms of OSs have been relatively well investigated, studies on their transformation are limited to a handful of studies on hydrolysis and heterogeneous oxidation, mechanisms and kinetics are not fully understood.

Ubiquitous in atmospheric aerosols, OSs have been considered relatively stable over their atmospheric lifetimes and thus used as tracers (Budisulistiorini et al., 2015). Nonetheless, experiments have shown that OSs are prone to transformation via hydrolysis and heterogeneous oxidation by OH. Elrod and co-workers recently reported that OSs can undergo hydrolysis to form polyols and sulfuric acid at rates subject to the aerosol acidity and molecular structure of particular OSs (Darer et al., 2011; Hu et al., 2011). In terms of the structural effects on OS properties and reactivity, they found that tertiary OSs readily undergo hydrolysis at relevant atmospheric aerosol pH (0−5) (Craig et al., 2018), which are reflected by their starkly shorter chemical lifetimes. Specifically, at pH = 0, the lifetimes of tertiary OSs range from only 0.1 day to 19 days, while those of primary and secondary OSs exceed 104 days against hydrolysis. In addition, Hu et al. (2011) discovered that the standard state free energies of hydrolysis for OSs investigated in their work were negative, implying that they are only metastable species and possibly not stable products.

Oxidation is another potential transformation pathway of OSs, as observed in previous work on the heterogeneous OH oxidation of OSs (Kwong et al., 2018; Lam et al., 2019; Chen et al., 2020; Xu et al., 2020a). Transformation of OSs, including methylsulfate, ethylsulfate, and isoprene-derived OSs proceeds at significant rates with a lifetime of about one to two weeks, and the proposed mechanism is that the investigated OSs can be fragmented into smaller products and sulfate radical anions ($SO_4^{\bullet-}$), which can participate in further reactions to form inorganic sulfate and other products. Furthermore, while over a hundred OSs have been detected in atmospheric aerosols, many of them are still unidentified, with unknown precursors and formation processes. Currently, reactions whose mechanisms are already understood fail to fully explain the formation of many OSs detected in atmospheric aerosols. Since it is possible for a variety of reaction products to be produced from heterogeneous reactions of OSs, some unidentified OSs (smaller than $C_5$) in atmospheric aerosols could be products generated upon further oxidation of OSs (e.g. 2-methyltetrol sulfates) formed through the photochemical reactions of VOCs (e.g. isoprene) (Chen et al., 2020). Altogether, previous laboratory findings have prompted the conjecture that the abundance of OSs reported in field studies may have been underestimated if their removal processes have not been properly accounted for. Another significant implication is the urgency to obtain better understanding of the transformation of OSs, which can allow a better assessment of the sources and environmental impacts of atmospheric OSs.

α-Pinene is an atmospherically important biogenic VOC, which can undergo photochemical oxidation to form secondary organic aerosols (SOA) (Kanakidou et al., 2005; Pye et al., 2010; Guenther et al., 2012). OSs have been found to be among important constituents of α-pinene derived SOA in chamber studies and ambient aerosols (Surratt et al., 2008; Stone et al., 2012; Ma et al., 2014). Field studies reported that a variety of α-pinene derived OSs can contribute 0.6–7.7 % of total sulfate in atmospheric aerosols (Huang et al., 2015, 2018; Wang et al., 2018, 2021). Here, we conducted a laboratory study to investigate the chemical transformation of a model α-pinene derived OS (αpOS-249, $C_{10}H_{17}O_5SNa$, Sodium 2-hydroxy-2,6,6-trimethylbicyclo [3.1.1] heptan-3-yl sulfate) through heterogeneous OH oxidation (**Table 1**). αpOS-249 can be formed through the photooxidation of α-pinene in the presence of acidic sulfate aerosols (Surratt et al., 2008). Oxidation experiments were performed using an oxidation flow reactor (OFR) at 50 % RH. After oxidation, aerosols were collected onto Teflon filters for chemical analysis. The composition of reaction products was characterized using the high-performance liquid chromatography-electrospray ionization-high resolution mass spectrometry and the ion chromatography. We first quantify oxidation kinetics by obtaining the effective heterogenous OH oxidation rate constant based on the decay of αpOS-249 against OH exposure. Second, possible transformation pathways are proposed to explain the formation of the detected products. In particular, we examine whether ambient OSs and inorganic sulfate could be produced upon heterogeneous OH oxidation of αpOS-249.

**Table 1**. Information of the α-pinene derived organosulfate investigated in this work.

| Name | αpOS-249 |
|---|---|
| Synonyms | Sodium 2-hydroxy-2,6,6-trimethylbicyclo[3.1.1] heptan-3-yl sulfate |
| Formula | $C_{10}H_{17}O_5SNa$ |
| Molecular weight (g/mol) | 272.29 |
| Chemical structure [a] |  |
| OH exposure ($\times 10^{11}$ molecule cm$^{-3}$ s) | 0−17.4 |
| Mean surface weighted diameter prior to oxidation (nm) | 181.3 ± 0.5 |
| Effective heterogeneous OH reaction rate constant, $k$ ($\times 10^{-13}$ cm$^3$ molecule$^{-1}$ s$^{-1}$) | 6.72 ± 0.55 |
| Atmospheric lifetime (days) | 11.5 ± 0.9 |

[a] Primary, secondary, and tertiary carbons are circled by red, blue, and green, respectively.

## 2. Experimental Method

### 2.1 Oxidation Experiments

αpOS-249 (in the form of its sodium salt) was synthesized by Yu and her co-workers (Wang et al., 2017) through the Upjohn dihydroxylation and sulfation of α-pinene. The purity (~99 %) has been tested using gas chromatography/electron impact-mass spectrometry, liquid chromatography/electrospray ionization-high resolution mass spectrometry and $^1$H and $^{13}$C nuclear magnetic resonance. Heterogeneous OH oxidation of αpOS-249 aerosols was carried out using a 13-L aluminium OFR at 50 ± 2.0 % RH and 298.0 ± 0.5 K. Experimental details together with schematic diagram (**Scheme S1**) are given in the Supporting Information. Briefly, αpOS-249 was first dissolved in deionized water (0.1 wt %) followed by a 30-min sonication. Aqueous aerosols were generated by passing the solution through an atomizer (TSI Model 3076) using 3 L min$^{-1}$ of nitrogen (N$_2$). The aerosol stream was then directly mixed with ozone (O$_3$), wet/humidified nitrogen (N$_2$) and oxygen (O$_2$) to control the RH. A total flow of ~5 L min$^{-1}$ was fed into the reactor, corresponding to a residence time of ~156 s (Xu et al., 2020a).

Gas-phase OH radicals were generated by photolyzing O$_3$ with UV light at 254nm in the presence of water vapor inside the OFR. The concentration of gas-phase OH radical was varied by changing the O$_3$ concentrations. The OH exposure, a product of gas-phase OH radical concentration and the residence time, was in the range of 0–17.4 × 10$^{11}$ molecule cm$^{-3}$ s. It was determined by measuring the decay of sulfur dioxide (SO$_2$) in independent calibrating experiments (Teledyne SO$_2$ analyzer, Model T100) based on the reaction rate between gas-phase OH radicals and SO$_2$ (= 9.0 × 10$^{-13}$ molecule$^{-1}$ cm$^3$ s$^{-1}$) at 298 K (Kang et al., 2007). It acknowledges that the presence of aerosols did not significantly affect the generation of gas-phase OH radicals and the determination of OH exposure (less than ~10 %). The aerosol stream leaving the reactor passed through an annular Carulite catalyst denuder (manganese dioxide/copper oxide catalyst; Carus Corp.) and an activated charcoal denuder to remove residual O$_3$ and other gas-phase species. Aerosols were collected onto the Teflon filters (47mm, 2.0 μm pore size, Pall Corporation) through filtration at a sampling flow rate of 3 L min$^{-1}$ using an air sampling pump (Gilian 500, Sensidyne) for 30 min, with a total gas sampling volume of ~ 90 L. Duplicate filters were collected from each of oxidation experiments for subsequent chemical analysis. After collection, filters were immediately stored at −20 °C in the dark and analysed within 3 months. Part of the remaining stream was introduced into a scanning mobility particle sizer (SMPS, TSI, CPC Model 3775, Classifier Model 3081) to measure the size distribution of the aerosols. The aerosol

mass was determined from measured volume concentration assumed for spherical aerosols with a unit density. Before oxidation, the mean surface weighted diameter for aerosol distribution was about $181.3 \pm 0.5$ nm with a geometric standard deviation of 1.3 and the aerosol mass loading was measured to be ~2000 $\mu g \ m^{-3}$.

## 2.2 Chemical Characterization of αpOS-249 and OSs formed upon Oxidation

**Scheme S2** shows the overview of the chemical analysis. First, the filters were extracted twice with 5 mL methanol in an ultrasonic bath for one hour. Extracts were then filtered through a 0.2 μm polytetrafluoroethylene (PTFE) syringe filter and combined. 300 μL of the extract was blown to dryness under a gentle stream of $N_2$ at room temperature and then reconstituted in 1 mL methanol–water (1:1 vol/vol) containing 200 ppb $D_{17}$-octyl sulfate as an internal standard.

**2.2.1 HPLC/ESI-QToF-MS**: To characterize the reaction products (i.e. OSs), 5 μL of reconstituted extract was injected into an Agilent 1290 UHPLC system equipped with an ESI source interfaced to an Agilent 6540 Quadrupole-Time-of-Flight Mass Spectrometer (HPLC/ESI-QToF-MS). Experimental details have been given elsewhere (Wang et al., 2017, 2021). Sample injections were first separated using an Acquity UPLC HSS T3 column (2.1 mm × 100 mm, 1.8 μm; Waters, Milford, MA) with mobile phase consisting of water ($H_2O$) (eluent A) and methanol (eluent B), each containing 0.1 % formic acid, at a flow rate of 0.3 mL $min^{-1}$. The gradient elution program was as follows: eluent B initially was set at 5 % for 2.0 min, increased to 95 % in 10.0 min, and held for 2 min; then decreased to 5 % in the next 0.1 min and held for 2.9 min. The ESI source was operated in the negative ion mode under following parameters: 2.8 kV for capillary voltage, 120 V for fragment, 320 °C for sheath gas temperature, 8 L $min^{-1}$ for drying gas flow and 45 psi for nebulizer pressure. Mass spectra were recorded across the range $m/z$ 50−1000 at 4 GHz with a resolution of 40,000 FWHM. The $MS^2$ spectra were also acquired at a collision energy of 13 eV to validate whether these ions are OSs by sulfur-containing fragments: $SO_3^{-\bullet}$ ($m/z$ = 79.9574), $HSO_3^-$ ($m/z$ = 80.9651), $SO_4^{-\bullet}$ ($m/z$ = 95.9523) and $HSO_4^-$ ($m/z$ = 96.9601) (Surratt et al., 2008; Hettiyadura et al., 2017). Data were analyzed using Mass Hunter Qualitative software (version B.07.00 Agilent Technologies).

**2.2.2 HPLC/ESI-QTRAP-MS**: To quantify the amount of αpOS-249 before and after oxidation, 5 μL of reconstituted extract was injected into an Agilent 1260 LC system (Palo Alto, CA) interfaced with a QTRAP 4500 mass spectrometer (AB Sciex, Toronto, Ontario,

Canada) and a TurboIonSpray source operated in multiple reaction monitoring (MRM) mode (Wang et al., 2017, 2021). LC separation was preformed using the same column and mobile phase adopted for HPLC/ESI-QToF-MS. The gradient elution program was set to be: eluent B initially was set at 1 % for 2.7 min, increased to 54 % in 17.9 min, and held for 1 min; then increased to 90 % in the next 7.5 min and held for 0.2 min; and finally decreased to 1 % in 1.8 min and held for 9.3 min. **Table S1** shows the parameters optimized for the mass transition between the deprotonated molecular ion of αpOS-249 ($C_{10}H_{17}O_5S^-$) and bisulfate ion ($HSO_4^-$). The transition was set to have a dwell time of 100 ms. The scan rate was 200 Da per s. Calibration curve was generated using αpOS-249 standard solution (retention time = 17.8 min) with 200 ppb $D_{17}$-octyl sulfate as an internal standard. The extraction efficiency of αpOS-249 was determined to be 85.3 ± 3.4 % by measuring the recovery of αpOS-249 standard spiked into blank filters. The uncertainty associated with the quantification of αpOS-249 is discussed in the Supporting Information.

## 2.3 Quantification of Inorganic Sulfate formed upon Oxidation

The amount of inorganic sulfate ($SO_4^{2-}$) formed upon oxidation was quantified using IC method. Operating conditions have been given by Huang et al. (2018). Briefly, the filters were extracted using 5 mL of double de-ionized water (18.2 MΩ cm) from the ultrapure water system (Nanopure Diamond UV/UF) and were sonicated for 60 min and then mechanically shaken for 60 min. After filtering, these extracts were subsequently analyzed by an ion chromatograph (Dionex ICS-1100). The separation of anions was accomplished using an AS11-HC analytical column (IonPac, $4 \times 250$ mm) and an AG11-HC guard column (IonPac, $4 \times 50$ mm) with 15 mmol $L^{-1}$ NaOH eluent. It is known that bisulfate ion ($HSO_4^-$) is being converted into $SO_4^{2-}$ upon mixing with the alkaline eluent. The concentration of $SO_4^{2-}$ quantified by the IC method thus represents a total amount of $HSO_4^-$ and $SO_4^{2-}$. Our pervious study has shown that the peak from the $Na_2SO_4$ standard has the same retention time in the IC chromatogram as that of the sodium bisulfate ($NaHSO_4$) standard (Xu et al., 2020a). Furthermore, the responses of the $NaHSO_4$ and $Na_2SO_4$ standards are about the same. These would justify the use of the $Na_2SO_4$ standard for the quantification of $HSO_4^-$ and $SO_4^{2-}$. In this work, the amount of $HSO_4^-$ and/or $SO_4^{2-}$ produced upon oxidation at a given OH exposure was proportional to its peak area in the chromatogram and was determined using the $Na_2SO_4$ standard calibration curve. The extraction efficiency was determined to be 90.3 ± 1.5 % by measuring the recovery of the $Na_2SO_4$ standard spiked onto blank filters. The uncertainty for the measurement of $SO_4^{2-}$ is discussed in the Supporting Information.

We also note that no distinct peak was detected in the ion chromatogram for αpOS-249 standard while a small peak corresponding to $SO_4^{2-}$ was observed. The absence of αpOS-249 peak could attribute to that IC detection for ionic and ionizable species is likely to be limited to small compounds (up to $C_5$) (Surratt et al., 2008; Domingos et al., 2012). The presence of $SO_4^{2-}$ peak may be due to the hydrolysis of αpOS-249. However, this peak comprised $2.1 \pm 0.8$ % of the total mass, suggesting that the hydrolysis of αpOS-249 was not significant. This observation is consistent with the literature that αpOS-249 does not readily undergo hydrolysis (Hu et al., 2015).

The high recovery of αpOS-249 suggests the sample preparation and extraction methods are effective. Wang et al. (2017) have also reported that there was no degradation for αpOS-249 after two-years' storage at low temperature (−20°C). A recent study by Hughes et al. (2019) examined the stability of a range of OSs (e.g. methyl sulfate, hydroxyacetone sulfate, two α-pinene derived OSs: $m/z = 279$ ($C_{10}H_{15}O_7S^-$), and $m/z = 281$ ($C_{10}H_{17}O_7S^-$)) on filters frozen at −20°C over one year. The filters were extracted via similar procedure applied in this study and the extracts were analyzed by HPLC-ESI-HRMS. They found that the investigated OSs with different functional groups (e.g. alkyl, carboxylate, and hydroxyl groups) showed no degradation during the storage. Taken together, αpOS-249 and its oxidation products (i.e. OSs) which have similar carbon skeletons while possessing different functional groups (alcohol and/or ketone) are likely stable during the storage and pre-treatment processes for chemical analysis.

We note that organic compounds such as carbonyls and carboxylic acids could undergo reactions with methanol during extraction, storage, and possibly during the electrospray process (Bateman et al., 2008). For instance, Batman et al (2008) suggested carboxylic acids could react with methanol to form esters and with carbonyls to hemiacetals and acetals. We checked the presence and relative abundance of these potential products (to that of our precursor, αpOS-249) in our aerosol mass spectra. At the maximum OH exposure, only a few products that could be potentially formed from the reactions of αpOS-249 with methanol were detected and they had negligible intensities. This would suggest that the influence of methanol is not significant on the identification of the major reaction products.

### 3. Results and Discussion

**Fig. 1** shows the total ion chromatogram (TIC) characterized by HPLC/ESI-QToF-MS for the OH oxidation of αpOS-249. Before oxidation (**Fig. 1a**), a single dominant peak corresponds to the deprotonated molecular ion ([M–H]$^−$) of αpOS-249 ($m/z$ = 249, $C_{10}H_{17}O_5S^−$). Upon oxidation at an OH exposure of $17.4 \times 10^{11}$ molecules cm$^{−3}$ s (**Fig. 1b**), αpOS-249 remains the dominant peak, accompanied by the appearance of some new product peaks in small intensity relative to αpOS-249. **Fig. 2** shows the extracted ion chromatograms (EICs) of the ions that are observed in the chromatograms after oxidation. Six product ions are detected upon oxidation and correspond to $m/z$ = 247 ($C_{10}H_{15}O_5S^−$), 263 ($C_{10}H_{15}O_6S^−$), 265 ($C_{10}H_{17}O_6S^−$), 277 ($C_{10}H_{13}O_7S^−$), 279 ($C_{10}H_{15}O_7S^−$), and 281 ($C_{10}H_{17}O_7S^−$). A mass tolerance was set to less than ± 5 ppm for assigning the chemical formula of the detected ions. On the basis of the chemical formulas, the ions that are detected are suggested to be OSs (**Table 2**). As shown in **Fig. S1**, MS$^2$ spectra show these ions fragmented into sulfur containing ions (SO$_3^{−•}$ ($m/z$ = 79.9574), HSO$_3^−$ ($m/z$ = 80.9651), and HSO$_4^−$ ($m/z$ = 96.9601)) and further confirmed the identity of these ions are OSs (Surratt et al., 2008; Hettiyadura et al., 2017; Wang et al., 2017, 2021). It may not be surprising that SO$_4^{−•}$ ($m/z$ = 95.9523) was not observed in these MS$^2$ spectra since it is likely originated from the fragmentation of tertiary OSs (Surratt et al., 2008).

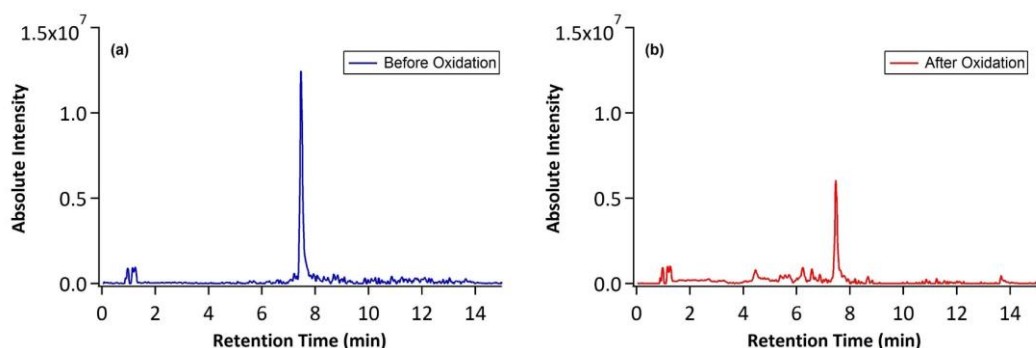

**Figure 1.** The total ion chromatograms (TIC) characterized by HPLC/ESI-QToF-MS before and after heterogeneous OH oxidation of αpOS-249.

Control experiments were also conducted to investigate the effects of O$_3$ and UV light on αpOS-249. **Fig. S2** shows that the impacts of UV photolysis and O$_3$ reactivity on αpOS-249 are not significant (less than 5 % change in the αpOS-249 signal in UV only and O$_3$ only experiments). Similar results have been observed for other OSs (e.g. methylsulfate, ethylsulfate, 2-methyltetrol sulfate, and 3-methyltetrol sulfate) that these OSs do not react with O$_3$ and photolyze at UV = 254 nm (Kwong et al., 2018; Lam et al., 2019; Chen et al., 2020; Xu et al., 2020a).

**Table 2**. Comparison of reaction products formed upon heterogeneous OH oxidation of αpOS-249 with the laboratory and field studies.

| Chemical formula [M-H]⁻ (theoretical mass) | This work | | Previous studies | | |
| --- | --- | --- | --- | --- | --- |
| | Retention time (min)[a] | Detected mass (error)[a] | Ambient mean concentration (ng m⁻³) | Laboratory experiment | Suggested chemical structure reported in the literature |
| $C_{10}H_{15}O_5S^-$ (247.0646) | 6.860 | 247.0648 (0.8095) | 3.32[b] | α-pinene OH/high-NOx/ highly acidic sulfate aerosols[d] | Not known |
| $C_{10}H_{15}O_6S^-$ (263.0595) | 4.992; 5.479; 5.787; 6.028; **6.227** | 263.0600 (1.9007) | 0.114[b] | Not observed | Not known |
| $C_{10}H_{17}O_6S^-$ (265.0751) | 3.283; 4.798; 5.420; 6.269; 6.408; **6.563**; 6.740 | 265.0762 (4.1498) | 0.076[b] | α-pinene /OH/high-NOx/ highly acidic sulfate aerosols[d]  Pinonaldehyde /acidic sulfate aerosols[e] |  |
| $C_{10}H_{13}O_7S^-$ (277.0387) | 6.117 | 277.0399 (4.3315) | Not observed | Not observed | Not known |
| $C_{10}H_{15}O_7S^-$ (279.0544) | 4.916; **5.384**; 5.551; 5.698 | 279.0545 (0.3584) | 7.1[c] | α-pinene /OH/high-NOx/ highly acidic sulfate aerosols[d] |  |
| $C_{10}H_{17}O_7S^-$ (281.0700) | 4.452 | 281.0709 (3.2020) | 12.1[c] | α-pinene /OH/high-NOx/ highly acidic sulfate aerosols[d] |  |

[a] The retention time in bold corresponds to the major peak in each EIC and only corresponding detected mass with mass errors (calculated in ppm) for these major peaks are listed for simplification. [b] These values were measured by the HPLC-triple quadrupole(TQ)-MS using sodium octyl sulfate as surrogate for quantification (Ma et al., 2014). [c] These values were measured by the HPLC- triple quadrupole(TQ)-MS using hydroxyacetone sulfate as surrogate for quantification (Hettiyadura et al., 2019). [d] Same ion has been detected in the α-pinene derived SOA formed from the photochemical oxidation of α-pinene in chamber studies (Surratt et al., 2008; Ma et al., 2014). [e] Same ion has been detected in the reactive uptake of pinonaldehyde on acidic sulfate aerosols (Liggio et al., 2006). [f] Chemical structure proposed by Liggio et al. (2006). [g] Chemical structure proposed by Surratt et al. (2008). [h] Chemical structure proposed by Hettiyadura et al. (2019).

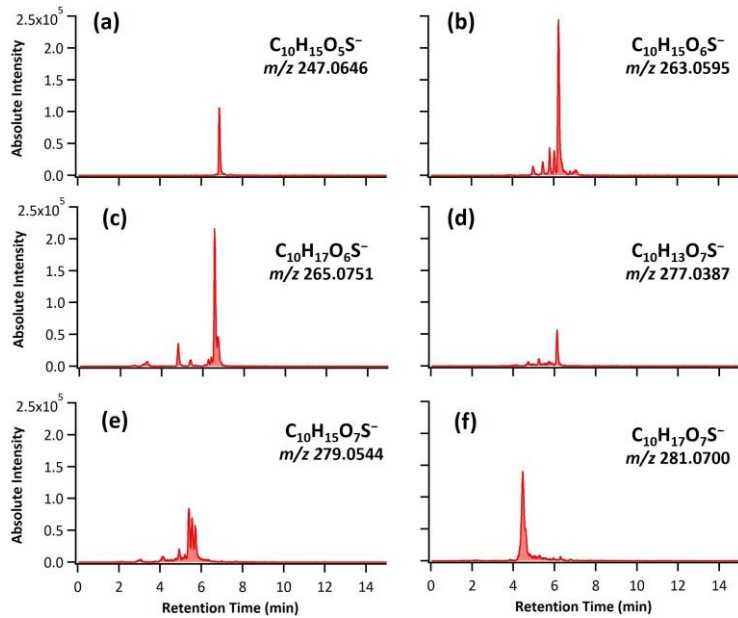

**Figure 2**. The extracted ion chromatograms (EICs) of the ions associated with reaction products formed upon heterogeneous OH oxidation of αpOS-249 at the highest OH exposure by HPLC/ESI-QToF-MS with mass tolerance of ± 5 ppm to their theoretical masses (shown).

### 3.1 Oxidation Kinetics

As shown in **Fig. 3**, αpOS-249 decays at a significant rate upon oxidation. Approximately 30 % of αpOS-249 remains unreacted at the highest OH exposure. Oxidation kinetics can be quantified by measuring the decay of αpOS-249 at different OH exposures. The normalized decay can be fit

with an exponential function (Smith et al., 2009):

$$ln \frac{I}{I_0} = -k\,[OH] \cdot t \qquad \text{(Eq. 1)}$$

where $I$ is the concentration of αpOS-249 quantified by HPLC/ESI-QTRAP-MS at a given

OH exposure, $I_0$ is the concentration before oxidation, $k$ is the effective second-order heterogeneous OH rate constant and $[OH] \cdot t$ is the OH exposure. The rate constant determined using **Eq. 1** is found to be $(6.72 \pm 0.55) \times 10^{-13}$ cm$^3$ molecule$^{-1}$ s$^{-1}$. Assuming a 24-h averaged [OH] of $1.5 \times 10^6$ molecules cm$^{-3}$ (Mao et al., 2009), the atmospheric lifetime of αpOS-249 against heterogeneous OH oxidation, $\tau = 1/k[OH]$ is calculated to be $11.5 \pm 0.9$ days. Considering atmospheric aerosols

with a similar size (~200 nm) having a typical lifetime of 10−14 days against wet/dry deposition (Kanakidou et al., 2005), heterogeneous OH oxidation could be a competitive sink for αpOS-249. Lam et al. (2018) and Chen et al. (2020) recently reported that isoprene-derived OSs (i.e. 2-methyltetrol sulfate and 3-methyltetrol sulfate) can undergo heterogeneous OH oxidation efficiently. Altogether, these results may suggest that transformation kinetics and pathways would need to be

considered in chemical transport models in order to better predict the abundance and composition of atmospheric OSs.

In this study, the heterogenous OH reactivity of aqueous αpOS-249 aerosols at a single RH (50 %) was investigated. In the atmosphere, the complex interplay between aerosol phase state (e.g. solid or aqueous), morphology and the types and concentrations of salts could significantly alter the heterogenous OH reaction kinetics and mechanisms under different environmental conditions (e.g. RH and temperature).

The aerosol physical state can play a key role in determining the heterogeneous kinetics and chemistry of pure organic aerosols (Koop et al., 2011; Shiraiwa et al., 2011, 2013; Chan et al., 2014). For instance, Chan et al. (2014) reported that aqueous succinic acid aerosols reacted about 40 time faster than in solid aerosols towards heterogeneous OH oxidation. These could be explained by the more rapid diffusion of succinic acid to the surface of aqueous droplets for oxidation than solid aerosols. Moreover, for aqueous droplets, aerosol water content can vary considerably, depending on atmospheric conditions. The change in aerosol-phase water and solutes concentrations would influence the reactivity by varying aerosol viscosity (Slade and Knopf, 2014; Chim et al., 2017; Marshall et al., 2016, 2018). For instance, oxidation kinetics in highly concentrated aqueous organic aerosols are found to be much slower than those in diluted ones. This is because aerosol viscosity generally increases with the solute concentration, thereby slowing down the diffusion of organic molecules within the aerosol and lowering the overall reactivity.

Atmospheric aerosols are comprised of organic compounds, inorganic salts and many other species. To date, large uncertainty remains in how inorganic salts alter the heterogeneous kinetics and chemistry (McNeill et al., 2007, 2008; Dennis-Smither et al., 2012). A few laboratory studies have revealed that the presence of dissolved inorganic ions (e.g., ammonium sulfate) can reduce the heterogeneous OH reactivity of organic compounds but does not significantly alter the reaction mechanisms (Mungall et al., 2017; Kwong et al., 2018; Lam et al., 2019). More recently, Xu et al. (2020b) reported that the change in the heterogeneous reactivity strongly depend on the concentration of organic compounds and inorganic salts. They found that the rate of the reactions decreases when the organic-to-inorganic mass ratio (OIR) decreases. This could be explained by the colliding probability between OH radical and organic species at the aerosol surface becoming lower in the presence of salt, resulting in a smaller overall reaction rate. We also acknowledge that different inorganic ions could have different propensity for air-aerosol interface depending on their polarizability and interactions with other components (Jungwirth and Tobias, 2002; Gopalakrishnan et

al., 2005). Jungwirth and Tobias (2002) investigated the preference of sodium cation and chlorine anion to the interface or bulk in sea salts aerosols based on polarizable MD simulation. They found that chlorine anion has a stronger propensity for the interface than sodium cation and is proportional to its polarizability. Therefore, the types and concentration of inorganic ions (e.g. ionic strength) within aerosol could potentially alter the overall heterogeneous kinetics and reaction pathways. To date, it remains an open question whether the salts alter heterogeneous reactivity chemically, physically, or both.

We would like to note that additional uncertainties in heterogeneous reactivity of organic compounds in the presence of inorganic salts could also arise when these organic–inorganic droplets undergo phase separation, depending on environmental conditions and aerosol composition (e.g. different types of inorganic salts, the average oxygen-to-carbon (O:C) elemental ratio of organic compounds, and OIR) (You et al., 2014; Qiu and Molinero, 2015; Freedman, 2017, 2020). These phase-separated droplets typically exhibit two distinct liquid phases: an inorganic-rich inner phase and an organic-rich outer phase. Different morphologies have also been observed (e.g. core–shell morphology, partially engulfed morphology and transitions between different types of morphology). Lam et al. (2021) recently reported that phase-separated organic–inorganic droplets have a slightly higher reactivity towards gas-phase OH radicals, compared to single-phase ones. As phase separation occurred, an uneven distribution of organic species within the droplets increased the collision probability between organic molecules and OH radicals at or near the droplet surface. Overall, further studies emphasized on the effects of aerosol composition, phase transition and separation, and morphologies on heterogeneous reactivity of OSs are needed to better understand their transformation rates and chemistry.

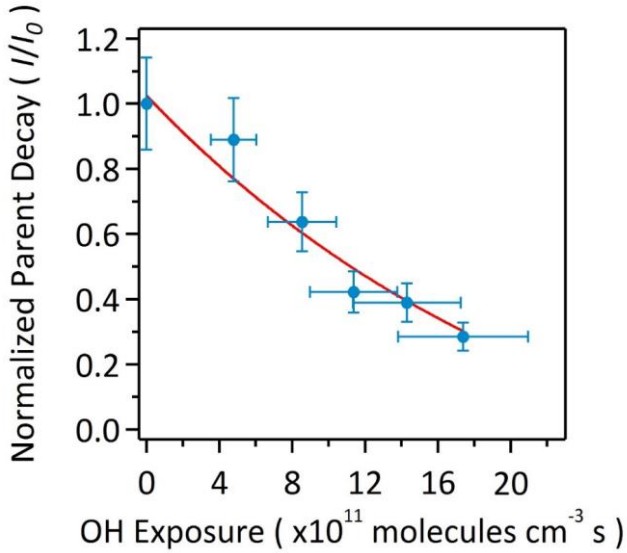

**Figure 3**. The normalized decay of αpOS-249 upon heterogeneous OH oxidation.

## 3.2 Reaction Mechanism

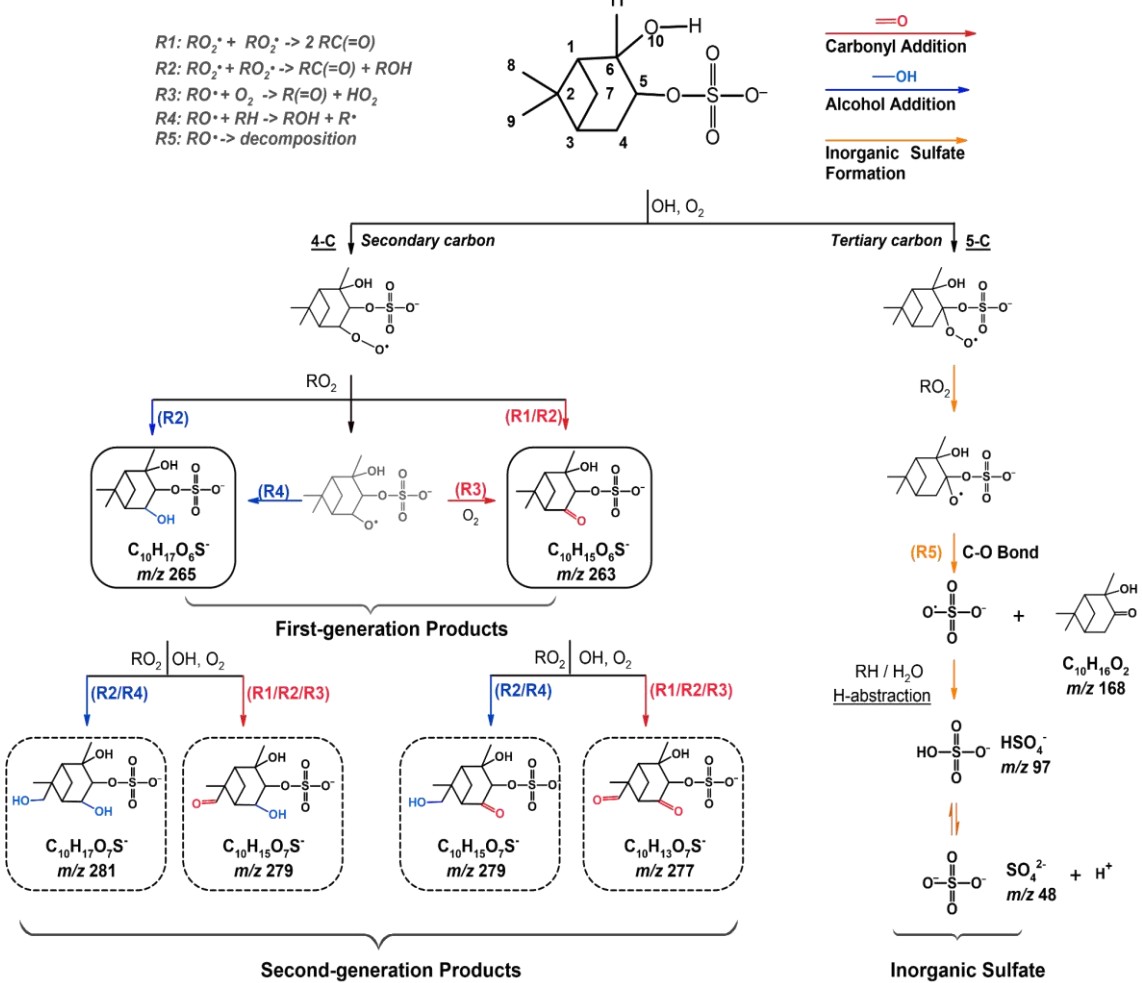

**Scheme 1**. Formation mechanisms tentatively proposed for the formation of more oxygenated $C_{10}$ OSs and inorganic sulfate upon heterogeneous OH oxidation of αpOS-249.

**Scheme 1** shows the proposed formation pathways for the detected products summarized in **Table 2**. Prior to oxidation, αpOS-249 tends to dissociate readily. Oxidation is initiated via hydrogen abstraction by an OH radical, forming an alkyl radical ($R\bullet$), which reacts with an oxygen ($O_2$) molecule quickly to form a peroxy radical ($RO_2\bullet$), which can react via different pathways. For instance, the reactions of two $RO_2\bullet$ can generate two carbonyl products via Bennett and Summers mechanism (**R1**) (Bennett and Summers, 1974), an alcohol product and a carbonyl product via Russell reactions (**R2**) (Russell, 1957), or two alkoxy radicals ($RO\bullet$). $RO\bullet$ can react with an $O_2$ molecule (**R3**), undergo intermolecular hydrogen abstraction (**R4**), and/or decompose involving the cleavage of a C-C bond or a C-O bond (George and Abbatt, 2008; Carrasquillo et al., 2015; Kroll et al., 2015).

$$RO_2\bullet + RO_2\bullet \rightarrow 2R(=O) \textbf{ (R1)}$$

$$RO_2\bullet + RO_2\bullet \rightarrow R(=O) + ROH \textbf{ (R2)}$$

$$RO_2\bullet + RO_2\bullet \rightarrow 2RO\bullet$$

$$RO\bullet + O_2 \rightarrow R(=O) + HO_2 \textbf{ (R3)}$$

$$RO\bullet + R'H \rightarrow ROH + R'\bullet \textbf{ (R4)}$$

$$RO\bullet \rightarrow decomposition \textbf{ (R5)}$$

Based on detected products (**Table 2** and **Fig. 2**), OH oxidation tends to increase the functionalities and oxygen content of αpOS-249. Moreover, these products ($C_{10}$ OSs) can be classified as functionalization products and are formed via the addition of one or two oxygenated functional groups to αpOS-249. For instance, ions corresponding to first-generation products produced from OH oxidation of αpOS-249 are detected at $m/z = 263$ ($C_{10}H_{15}O_6S^-$) and 265 ($C_{10}H_{17}O_6S^-$). When the oxidation proceeds, these products can react with OH radicals to form second-generation products, which give $m/z = 277$ ($C_{10}H_{13}O_7S^-$), 279 ($C_{10}H_{15}O_7S^-$), and 281 ($C_{10}H_{17}O_7S^-$) products.

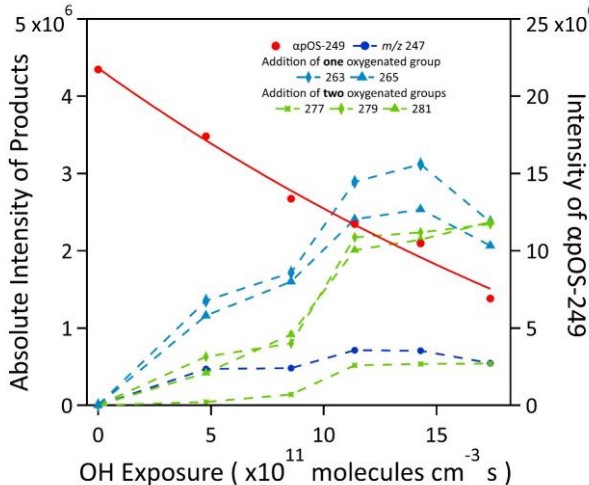

**Figure 4**. The evolution in the signal intensity of the reaction products formed upon the heterogeneous OH oxidation of αpOS-249 as a function of OH exposure.

**Fig. 4** shows the evolution of the intensity of the reaction products as a function of OH exposure. Different kinetic profiles have been observed for first- and second-generation products. It can be seen that first-generation products ($m/z = 263$ ($C_{10}H_{15}O_6S^-$) and 265 ($C_{10}H_{17}O_6S^-$)) reach a maximum intensity at an OH exposure of about $14.3 \times 10^{11}$ molecules $cm^{-3}$ s. Their intensities then decrease slightly at the highest OH exposure. The concentration of second-generation products ($m/z = 277$ ($C_{10}H_{13}O_7S^-$), 279 ($C_{10}H_{15}O_7S^-$), and 281 ($C_{10}H_{17}O_7S^-$)) follows a different kinetic profile. At low OH exposures, the intensities of the products are observed to increase at a slightly slower rate relative to first-generation products. Their concentrations always increase with increasing OH exposure and reach their maximum values at the highest OH exposure. This trend is consistent with the multigenerational reactions that the formation of second-generation products is from the reactions of OH with first-generation products. In general, the products detected, shown in **Table 2**, are

consistent with reactions (described in detail below) that form hydroxyl and carbonyl functional groups.

During oxidation, the carbon site where the hydrogen atom is abstracted by OH radical governs the following formation of products. More recently, molecular dynamics (MD) simulations showed that heterogeneous reaction is likely not initiated by the direct collision between a gas-phase OH radical and an organic molecule near the aerosol surface (Xu et al., 2020b; Lam et al., 2021). Instead, the reaction may occur after a number of collisions between the absorbed OH radical and the organic molecule. As a first approximation, the reactivity of αpOS-249 towards OH radical is estimated by a structure–activity relationship (SAR) model proposed for aqueous-phase OH reactions with organic compounds (Monod and Doussin, 2008). We acknowledge that the SAR model does not include the parameterization of the sulfate group ($-OSO_3^-$) for OSs. As the sulfate group ($-OSO_3^-$) exhibits a resonance electron-withdrawing effect (Bahl et al., 2012) as the carboxylate anion ($-COO^-$) and bears a negative charge, here the effect of the sulfate group on the OH reactivity is evaluated using the descriptor of the carboxylate anion ($-COO^-$) in the SAR model. Overall, the goal of this simple analysis is to qualitatively assess which carbon site is more favorable for hydrogen abstraction in reactions of OH radical with αpOS-249 in order to gain more insights into the reaction mechanisms.

**Table S2** shows the reactivity of hydrogen atom at different reaction sites upon OH oxidation of αpOS-249. The model predicts that the hydrogen abstraction is likely occurred at two secondary carbons (4-C and 7-C) and a tertiary carbon (3-C), in which the abstraction rates were significantly larger than other sites. For instance, the largest hydrogen abstraction rate is predicted to occur at a secondary carbon site (4-C). This could be explained by the electron donor effect of the carboxylate anion ($-COO^-$) group in β-position when the OH attacks the reactive site as an electrophilic reaction (Monod and Doussin, 2008). A comparable rate is predicted for a tertiary carbon (3-C) as the adjacent α-CH$_2$ and β-CH$_3$ groups exhibit significant electron donor properties. A large abstraction rate for 7-C could also be due to strong electron-donating effect of surrounding α-CH groups. The model suggests a large difference in the OH reactivity for the three tertiary carbon sites (1-C, 3-C, and 5-C). For instance, the slowest abstraction rate is predicted for 5-C. This could attribute to a combined effect of the resonance electron-withdrawing effect of the α-carboxylate anion ($-COO^-$) group and the field electron withdrawing effect of the β-OH group. The slow abstraction rate predicted for 1-C could be due to the field electron withdrawing effect of the β-OH group. In general, the OH reactivity for the primary carbons (8-C, 9-C, and 11-C) and hydroxyl group (10-O) were predicted to be slower than that of the secondary and tertiary carbons except 5-C. We would like to acknowledge that uncertainties could arise from that the sulfate group ($-OSO_3^-$) is currently not being considered in the SAR model and differences between the concentrated aqueous aerosols and

the SAR model that is formulated for dilute aqueous solutions. Other factors such as steric hindrance and long-distance electronic effects of the sulfate group ($-OSO_3^-$) on the reactivity have not been considered (Yamazaki et al., 2019) and are warranted for further studies.

Depending on the initial reaction site, a number of first-generation products with various structural isomers can be formed as indicated by multiple peaks observed in the EICs of detected ions (**Fig. 2**). Different isomers of second-generation products can also be formed from the reactions of OH with first-generation products. The possible formation pathways of detected products are further discussed below. Given a number of reaction pathways could possibly lead to the for-
mation of the products with different isomers, only a general reaction scheme is depicted in **Scheme 1** for simplicity and clarity.

### 3.2.1 First-generation products

$m/z = 263$ ($C_{10}H_{15}O_6S^-$): Upon oxidation, $C_{10}H_{15}O_6S^-$ is likely formed via an addition of a
carbonyl functional group to αpOS-249 ($m/z = 249$, $C_{10}H_{17}O_5S^-$) after the hydrogen abstraction by an OH radical from a primary or a secondary carbon instead of a tertiary carbon (**Scheme 1**). Depending on different initial reaction sites (4-C, 7-C, 8-C, 9-C, and 11-C), five structural isomers of $C_{10}H_{15}O_6S^-$ could be possibly formed from the cross-reactions of two $RO_2\bullet$ via Bennett and Summers reactions (**R1**), Russell reactions (**R2**) and/or the reactions between an alkoxy radical with an
$O_2$ molecule (**R3**).

$m/z = 265$ ($C_{10}H_{17}O_6S^-$): $C_{10}H_{17}O_6S^-$ is likely formed via an addition of a hydroxyl functional group to αpOS-249 ($m/z = 249$, $C_{10}H_{17}O_5S^-$) upon oxidation as shown in **Scheme 1.** The addition of a hydroxyl functional group to αpOS-249 can possibly occur at all carbons, except the two qua-
ternary carbons (2-C and 6-C) where there is no hydrogen atom available for abstraction. A total of eight isomers could be possibly formed from the cross-reactions of two $RO_2\bullet$ via Russell reactions (**R2**), and the intermolecular hydrogen abstraction by alkoxy radicals (**R4**).

We also note that a product ion at $m/z = 247$ ($C_{10}H_{15}O_5S^-$) was detected after oxidation
(**Fig. 2**). One possibility is that this product might be originated from the transformation of $C_{10}H_{17}O_6S^-$ ($m/z = 265$). **Scheme S3 (Path a)** shows that when the hydrogen abstraction occurs at the 1-C, the Pinacol rearrangement of vicinal diols involving a methyl shift and a loss of $H_2O$ could be a possible formation pathway (Bruice et al., 2004). This reaction occurs efficiently under acidic conditions and might not be favorable in this study since αpOS-249 aerosols are expected to be
neutral. Another possible pathway involves the isomerization of a hydroxyl alkyl radical with a

ring opening, as shown in **Scheme S3 (Path b)**. This reaction has been found to be efficient for gas-phase reactions between α-pinene and OH radicals (Bergh et al., 2000; Peeters et al., 2001). Subsequent reactions of the hydroxyl alkyl radical could lead to the formation of $m/z = 247$ ($C_{10}H_{15}O_5S^-$) through the $HO_2$ elimination processes. However, the significance of this reaction in aerosol-phase requires further investigation.

### 3.2.2 Second-generation products

As shown in **Fig. 2**, second-generation products with different isomers are formed through an addition of a carbonyl or a hydroxyl functional group to first-generation products upon oxidation. For instance, as shown in **Scheme 1**, $C_{10}H_{17}O_7S^-$ ($m/z = 281$) can be formed from the further oxidation of $C_{10}H_{17}O_6S^-$ ($m/z = 265$) with an addition of a hydroxyl functional group via **R2** and **R4**. $C_{10}H_{13}O_7S^-$ ($m/z = 277$) can be formed from the oxidation of $C_{10}H_{15}O_6S^-$ ($m/z = 263$) with an addition of a carbonyl functional group via **R1**-**R3**. Multiple pathways could potentially lead to the formation of $C_{10}H_{15}O_7S^-$ ($m/z = 279$). For instance, $C_{10}H_{15}O_7S^-$ ($m/z = 279$) can be formed through the addition of a hydroxyl functional group to $C_{10}H_{15}O_6S^-$ ($m/z = 263$) via **R2** and **R4** and/or the addition of a carbonyl functional group to $C_{10}H_{17}O_6S^-$ ($m/z = 265$) via **R1**-**R3**.

We note that smaller products and small OSs (C<10) were not observed in the mass spectra and ion chromatograms. We do not have a clear explanation yet, but postulate that upon OH oxidation of αpOS-249, the self/cross-reactions of peroxy radicals with a cyclic structure tend to form stable products rather than alkoxy radicals. For instance, in a theoretical study by Capouet et al. (2004), a smaller branching ratio (0.3) was assigned to alkoxy radical formation in the disproportionation reactions of cyclic peroxy radicals (0.7 is assigned for acyclic peroxy radicals) to reproduce the formation of reaction products during the photooxidation of α-pinene by OH radical. Rowley et al. (1992) also reported a low value (0.24) for alkoxy radical formation from the self-reactions of cyclohexylperoxy radicals. Another possible explanation is that a higher activation energy is estimated for the decomposition of an alkoxy radical when it is attached to larger ring structures (Wilsey et al., 1999). Altogether, the formation and decomposition of alkoxy radicals might be likely hindered, which may partly explain the insignificant formation of fragmentation products including small OSs (C <10) and inorganic sulfates (**Sect. 3.4**) upon OH oxidation of αpOS-249. This might also suggest that more oxygenated $C_{10}$ OSs ($C_{10}H_{15}O_7S^-$ ($m/z = 279$) and $C_{10}H_{17}O_7S^-$ ($m/z = 281$)) are more likely originated from further OH oxidation of first-generation products (**Scheme 1**) without fragmentation processes. For instance, as shown in **Scheme S4**, $C_{10}H_{15}O_7S^-$ ($m/z = 279$) and $C_{10}H_{17}O_7S^-$ ($m/z = 281$) could be formed through a C-C bond cleavage of RO• with a ring opening. The resulted carbonyl alkyl radical could undergo subsequent reactions, leading to the formation of $C_{10}H_{15}O_7S^-$ ($m/z = 279$) via the addition of a carbonyl group and $C_{10}H_{17}O_7S^-$ ($m/z = 281$) via the addition of a hydroxyl group. However, these fragmentation reactions are expected to be not significant. Further laboratory and modeling investigations are required to better

understand the fates of peroxy and alkoxy radicals with varying structures in different phases (i.e. gas phase vs. aerosol phase) and subsequent formation of reaction products (e.g. smaller OSs) upon oxidation.

## 3.3 Heterogeneous OH Oxidation of αpOS-249 - A Potential Source of Ambient OSs?

On the basis of reaction products, we attempt to examine whether the heterogeneous OH oxidation of αpOS-249 freshly formed from the photooxidation of α-pinene could explain the formation of OSs detected in laboratory generated α-pinene SOA and ambient aerosols. As shown in **Table 2**, five out of six product ions ($m/z$ = 247, 263, 265, 279, and 281) have been detected in ambient aerosols (Iinuma et al., 2005; Surratt et al., 2008; Ma et al., 2014; Hettiyadura et al., 2019; Wang et al., 2021) while four out of six product ions ($m/z$ = 247, 265, 279, and 281) have been observed for SOA formed from the photooxidation of α-pinene in the presence of acidic sulfate aerosols in laboratory studies (Liggio et al., 2006; Surratt et al., 2008; Ma et al., 2014; Zhang et al., 2015; Hettiyadura et al., 2019). Reaction mechanisms have been proposed for the formation of some of these products such as $m/z$ = 265 ($C_{10}H_{17}O_6S^-$) and 279 ($C_{10}H_{15}O_7S^-$) in the literature. For instance, in chamber studies, Liggio et al. (2006) suggested that $C_{10}H_{17}O_6S^-$ ($m/z$ = 265) can be formed through the reactive uptake of pinonaldehyde, a major semi-volatile product generated from α-pinene oxidation, onto acidic sulfate aerosols via sulfate esterification reactions. Similar to $C_{10}H_{17}O_6S^-$ ($m/z$ = 265), $C_{10}H_{15}O_7S^-$ ($m/z$ = 279) can be produced through the reactive uptake of hydroxypinoinc acid on acidic sulfate aerosols via sulfate esterification reactions (Surratt et al., 2008). For field studies, ions corresponding to $m/z$ = 247 ($C_{10}H_{15}O_5S^-$) and 263 ($C_{10}H_{15}O_6S^-$) have been observed in ambient aerosols, but their sources and formation mechanisms have remained unclear. Here, we show that these OSs previously observed in ambient aerosols and α-pinene derived SOA in laboratory studies could be first- and/or second-generation products formed upon heterogeneous OH oxidation of αpOS-249.

## 3.4 Formation of Inorganic Sulfate upon Oxidation

Recent studies have reported that heterogeneous OH oxidation of small OSs ($C_1$, $C_2$, and $C_5$) can produce inorganic sulfates (Kwong et al., 2018; Lam et al., 2019; Chen et al., 2020; Xu et al., 2020a). It has been proposed that the decomposition of an alkoxy radical formed at the α-position of the sulfate group can generate a sulfate radical anion ($SO_4^{\bullet-}$), which subsequently reacts further to generate inorganic sulfur species ($HSO_4^-$ and $SO_4^{2-}$) (**Scheme 1** and **Scheme S4**). We here examine the significance of this conversion from organosulfur to inorganic sulfur upon heterogeneous OH oxidation of αpOS-249 (a $C_{10}$ OS). **Fig. S3** shows the IC chromatograms of αpOS-249 before and after heterogenous OH oxidation. Before oxidation (**Fig. S3a**), a small quantity of $SO_4^{2-}$

(contributing only 1.9 ± 1.3 % of total sulfur mass) was detected in the ion chromatogram due to the hydrolysis of αpOS-249, which has been corrected for the determination of sulfate yield. After oxidation (**Fig. S3b**), an increase in the $SO_4^{2-}$ signal was observed, suggesting that some sulfur is being converted from its organic form (i.e. αpOS-249) into its inorganic form ($HSO_4^-$ and/or $SO_4^{2-}$) upon OH oxidation. The amount of inorganic sulfur formed from oxidation was then quantified to calculate the yield, defined as the total number of moles of $HSO_4^-$ and $SO_4^{2-}$ formed per mole of αpOS-249 reacted at a given OH exposure (Xu et al., 2020a):

$$\text{Yield} = \frac{\Delta \left[SO_4^{2-}\right]}{\Delta \left[\text{αpOS} - 249\right]} \times 100\%$$

where the $SO_4^{2-}$ concentration was quantified by IC and the αpOS-249 concentration was determined by HPLC/ESI-QTRAP-MS before and after oxidation. As shown in **Fig. 5**, the yields are determined to be small as indicated by the small $SO_4^{2-}$ peak detected after oxidation and range from 3.5 ± 11.3 % at OH exposure of $4.8 \times 10^{11}$ molecules cm$^{-3}$ s to 13.8 ± 4.7 % at $14.3 \times 10^{11}$ molecules cm$^{-3}$ s. The relatively large uncertainty at low OH exposures is mainly attributed to the small change in αpOS-249 and $SO_4^{2-}$ concentrations at low OH exposures (The determination of uncertainties of the yield is given in Supporting Information). The small yields could be explained by that the hydrogen abstraction at 5-C is likely not favorable compared to other carbon sites upon OH oxidation of αpOS-249 at the first place as predicted by the SAR model (**Table S2**). The generation of sulfate radical anions ($SO_4^{\bullet -}$) via the breakage of the C-O bond in 5-C alkoxy radical and the further reaction of $SO_4^{\bullet -}$ to generate inorganic sulfates might not be favorable (**Scheme 1 and Scheme S4**). Furthermore, we cannot rule out the possibility of the formation of $SO_4^{2-}$ could be from the hydrolysis of reaction products as some secondary and tertiary OS could readily undergo hydrolysis and may form upon oxidation.

It is also worthwhile to note that upon oxidation, the decomposition of alkoxy radicals (RO•) formed at 5-C would lead to $SO_4^{\bullet -}$ and a fragmentation product. For instance, a $C_{10}$ aldehyde product ($C_{10}H_{16}O_2$, $m/z = 167$) can be formed (**Scheme 1**). The effective saturation vapor pressure, $C^*$ of this product is estimated to be $2.27 \times 10^5$ μg m$^{-3}$ based on its saturation vapor pressure (3.29 Pa) predicted from EVAPORATION (Compernolle et al., 2011). The volatility of this product is found to be about 5 orders of magnitude larger than that of αpOS-249 ($C^* = 1.67$ μg m$^{-3}$), which is estimated using a saturation vapor pressure ($7.96 \times 10^{-6}$ Pa) predicted by COSMOtherm (Hyttinen et al. 2020). Given its high volatility, this fragmentation product likely partitions back to the gas phase and has not been detected in our chemical analysis.

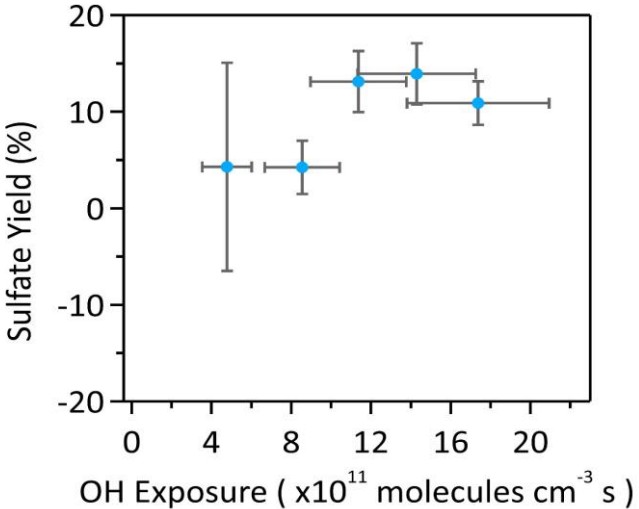

**Figure 5**. Molar yields of inorganic sulfur species ($HSO_4^-$ and $SO_4^{2-}$) as a function of OH exposure upon heterogeneous OH oxidation of αpOS-249. The x-error bars represent the uncertainties for OH exposures and y-error bars represent the uncertainties of derived molar yields (calculation details can be referred to Supporting Information).

### 4. Atmospheric Implications

To date, while the formation mechanisms of α-pinene derived OSs have been investigated, its transformation pathways have remained unclear. Nonetheless, several laboratory studies have documented the chemical removal pathways of several OSs through oxidation, implying that certain OSs are not chemically stable and have the propensity for transformation to other organic or inorganic species via oxidation. In this work, we investigated the oxidative transformation process of αpOS-249 via heterogeneous oxidation initiated by OH radicals. Kinetic data suggest that heterogeneous OH oxidation is a competitive sink for αpOS-249. Oxidation reactions dominantly involve the addition of hydroxyl and carbonyl functional groups, which increases the functionalities and oxygen content of αpOS-249 during multigenerational oxidation steps. The formation of fragmentation products such as inorganic sulfates and smaller OSs was found to be insignificant. All these results indicate that αpOS-249s and other OSs may not be chemically stable in the atmosphere and can be continuously transformed once formed in the atmosphere. In particular, heterogeneous OH oxidation of OSs could yield some previously unexplained OSs that are detected in atmospheric aerosols. We also acknowledge that the small yields reported for αpOS-249 are different from our previous studies that a significant amount of inorganic sulfates (yield ~ 60 %) was formed upon heterogeneous OH oxidation of a small OS (i.e. methylsulfate, $CH_3SO_4^-$) (Kwong et al., 2018; Xu et al., 2020a). This might be attributed to that for the OH reaction with methylsulfate, only a few hydrogen atoms are available for the abstraction. The formation and decomposition of an alkoxy radical formed at the α-position of the sulfate group are more likely occurred, leading to sulfate radical anion ($SO_4^{\bullet-}$) and subsequently inorganic sulfates. These results reveal that the sulfur

conversion from its organic form (i.e. OSs) to inorganic form (i.e. inorganic sulfates) upon oxidation could be sensitive to the molecular structure of OSs (e.g. carbon chain length and functionality). Future investigations are needed to better elucidate the formation and isomer distribution of multi-generational functionalization and fragmentation products formed upon oxidation. Lastly, smaller products and smaller OSs (C<10) were not detected upon oxidation of αpOS-249. Future investigations are desired to investigate whether the chemical transformation and ageing of larger OSs could be the sources of smaller OSs in the atmosphere through various atmospheric processes. Overall, the findings of this work provide an improved understanding of the sources, fates and transformations of ambient OSs.

**Data availability.** Data are available upon request from the corresponding author.

**Author contributions.** Rongshuang Xu and Man Nin Chan designed the experiments. Rongshuang Xu and Sze In Madeleine Ng ran the experiments. Yuchen Wang provided the synthesized αpOS-249 standard. Wing Sze Chow, Yee Ka Wong, Zhongping Yao, Pui-Kin So, and Jian Zhen Yu helped with the chemical analysis. Yuchen Wang and Jian Zhen Yu contributed to the formulation of the reaction mechanisms. Rongshuang Xu, Sze In Madeleine Ng, and Man Nin Chan prepared the manuscript. Jian Zhen Yu and Man Nin Chan edited the manuscript. All authors provided comments and suggestions for the manuscript.

**Competing interests.** The contact authors have declared that neither they nor their co-authors have any competing interests.

**Acknowledgements.** This work is supported by the Hong Kong Research Grants Council (14300118 and 16304519).

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
