# Peer review of "Chemical Transformation of $\alpha$ -Pinene derived Organosulfate via Heterogeneous OH Oxidation: Implications for Sources and Environmental Fates of Atmospheric Organosulfates"

_Atmospheric Chemistry and Physics, 2021_

## Referee Comment (RC1)

The work by Xu et al. investigated the heterogeneous OH oxidation of one α-Pinene derived organosulfate (i.e., $C_{10}H_{17}O_5SNa$, αpOS-249), and both reaction kinetics and mechanisms are well studied and discussed. This is an interesting work that uses a structure–activity relationship (SAR) to assess which carbon site is more favorable for hydrogen abstraction in reactions of OH radical with αpOS-249, further confirmed by the products measured. The experiments, model predictions, results, and conclusions are sound and well described. Therefore, this work could be published nearly as is. Before the publish, there are several minor comments the authors may consider.

General comments

1. Methanol is a common solvent used for MS analysis. However, it should be careful that many SOA constituents such as carbonyls and carboxylic acids undergo chemical reactions with methanol during extraction, storage, and possibly during the electrospray process (Bateman et al., 2008). The influence of methanol may be little on the results of this study, but it is better to give some explanations here.

Specific comments

1. Page 2 Line 16: Some supported references were suggested to be added here.
2. Page 5 Line 25: were → was
3. Page 18 Lines 13-14: Is it possible that inorganic sulfate was formed from the hydrolysis of organic products instead of a direct formation?

Bateman, A. P., Walser, M. L., Desyaterik, Y., Laskin, J., Laskin, A. and Nizkorodov, S. A.: The effect of solvent on the analysis of secondary organic aerosol using electrospray ionization mass spectrometry, Environ. Sci. Technol., 42(19), 7341–7346, doi:10.1021/es801226w, 2008.

---

## Author Comment (AC1)

*The work by Xu et al. investigated the heterogeneous OH oxidation of one α-Pinene derived organosulfate (i.e., $C_{10}H_{17}O_5SNa$, αpOS-249), and both reaction kinetics and mechanisms are well studied and discussed. This is an interesting work that uses a structure–activity relationship (SAR) to assess which carbon site is more favorable for hydrogen abstraction in reactions of OH radical with αpOS-249, further confirmed by the products measured. The experiments, model predictions, results, and conclusions are sound and well described. Therefore, this work could be published nearly as is. Before the publish, there are several minor comments the authors may consider.*

**We thank the reviewer for his/her thoughtful comments. The referee's comments are below in italics followed by our responses in normal font.**

**General Comment:**

*Methanol is a common solvent used for MS analysis. However, it should be careful that many SOA constituents such as carbonyls and carboxylic acids undergo chemical reactions with methanol during extraction, storage, and possibly during the electrospray process (Bateman et al., 2008). The influence of methanol may be little on the results of this study, but it is better to give some explanations here.*

*Bateman, A. P., Walser, M. L., Desyaterik, Y., Laskin, J., Laskin, A. and Nizkorodov, S. A.: The effect of solvent on the analysis of secondary organic aerosol using electrospray ionization mass spectrometry, Environ. Sci. Technol., 42(19), 7341–7346, doi:10.1021/es801226w, 2008.*

**Author Response:**

Thanks for this insightful comment. We agree with the reviewer's comment that organic compounds such as carbonyls and carboxylic acids could undergo reactions with methanol during extraction, storage, and possibly during the electrospray process. For instance, as suggested in the literature, carboxylic acids could react with methanol to form esters and with carbonyls to hemiacetals and acetals. This would result in the *m/z* shifts in the mass spectra: 14.0156 ($+CH_3OH-H_2O$), 32.0262 ($+CH_3OH$), 46.0419 ($+2CH_3OH-H_2O$) (Bateman et al., 2008). Based on our proposed reaction pathways, we checked the presence and relative abundance of these potential reaction products (to that of our precursor, αpOS-249) in our mass spectra. At the maximum OH exposure, only a few products that could be potentially formed from the reactions of αpOS-249 with methanol were detected and they had negligible intensities (Please see the table below). This would suggest the influence of methanol is not significant on the identification of the major reaction products in our study.

| Precursor | | Theoretical mass | |
|---|---|---|---|
| αpOS-249, $C_{10}H_{17}O_5S^-$ | | 249.0797 | |
| *m/z* shift | Formula | Theoretical mass | Relative abundance |
| 14.0156 | $C_{11}H_{19}O_5S^-$ | 263.0953 | 0.06% |
| 32.0262 | $C_{11}H_{21}O_6S^-$ | 281.1059 | 0.09% |
| 46.0419 | $C_{12}H_{23}O_6S^-$ | 295.1216 | 0.01% |

We have added the following information in the revised manuscript.

Page 8, Line 23: "We note that organic compounds such as carbonyls and carboxylic acids could undergo reactions with methanol during extraction, storage, and possibly during the electrospray process (Bateman et al., 2008). For instance, Batman et al (2008) suggested carboxylic acids could react with methanol to form esters and with carbonyls to hemiacetals and acetals. We checked the presence

and relative abundance of these potential products (to that of our precursor, αpOS-249) in our aerosol mass spectra. At the maximum OH exposure, only a few products that could be potentially formed from the reactions of αpOS-249 with methanol were detected and they had negligible intensities. This would suggest that the influence of methanol is not significant on the identification of the major reaction products."

The literature was also added into the reference list.

- Bateman, A. P., Walser, M. L., Desyaterik, Y., Laskin, J., Laskin, A. and Nizkorodov, S. A.: The effect of solvent on the analysis of secondary organic aerosol using electrospray ionization mass spectrometry, Environ. Sci. Technol., 42, 7341–7346, doi:10.1021/es801226w, 2008.

**Specific Comment #1:**
*Page 2 Line 16: Some supported references were suggested to be added here.*

**Author Response:**
Thanks for the suggestion. We have added supported references in the manuscript and the references.
Page 2, Line 17: "Sulfur-containing aerosols are of particular significance for human health because of their high abundance and significant impacts on regional air quality and global climate (Bentley et al., 2004; Riva et al., 2015; Stadtler et al., 2018)."

Following references were also added into the references:

- Bentley, R.; Chasteen, T. G.: Environmental VOSCs – formation and degradation of dimethyl sulfide, methanethiol and related materials, Chemos., 55, 291−317, https://doi.org/10.1016/j.chemosphere.2003.12.017, 2004.
- Riva, M.; Tomaz, S.; Cui, T. Q.; Lin, Y. H.; Perraudin, E.; Gold, A.; Stone, E. A.; Villenave, E.; Surratt, J. D.: Evidence for an Unrecognized Secondary Anthropogenic Source of Organosulfates and Sulfonates: Gas-Phase Oxidation of Polycyclic Aromatic Hydrocarbons in the Presence of Sulfate Aerosol, Environ. Sci. Technol, 49, 6654−6664, https://doi.org/10.1021/acs.est.5b00836, 2015.
- Stadtler, S.; Kühn, T.; Schröder, S.; Taraborrelli, D.; Schultz, M. G.; Kokkola, H.: Isoprene derived secondary organic aerosol in a global aerosol chemistry climate model, Geosci. Model Dev., 11, 3235−3260, https://doi.org/10.5194/gmd-11-3235-2018, 2018.

**Specific Comment #2:**
*Page 5 Line 25: were → was*

**Author Response:**
We have revised the sentence in the manuscript.
Page 5, Line 33: "Part of the remaining stream was introduced into a scanning mobility particle sizer (SMPS, TSI, CPC Model 3775, Classifier Model 3081) to measure the size distribution of the aerosols."

**Specific Comment #3:**
*Page 18 Lines 13-14: Is it possible that inorganic sulfate was formed from the hydrolysis of organic products instead of a direct formation?*

**Author Response:**
We agree with the reviewer that inorganic sulfate could also be generated from the hydrolysis of reaction products. This is because secondary and tertiary OS could readily undergo hydrolysis and may form upon OH oxidation based on our proposed reaction mechanisms. We have added this information in the manuscript.
Page 20, Line 22: "Furthermore, we cannot rule out the possibility of the formation of $SO_4^{2-}$ could be from the hydrolysis of reaction products as some secondary and tertiary OS could readily undergo hydrolysis and may form upon oxidation."

---

## Author Comment (AC2)

*Xu et al. presents laboratory studies where they investigated the heterogenous oxidation of an alpha-pinene organosulfate surrogate, sodium 2-hydroxy-2,6,6-trimethylbicyclco[3.1.1]heptan-3-yl sulfate, or apOS-249, using an oxidation flow reactor. The reaction and products were measured with an UHPLC-ESI-QToFMS, HPLC-ESI-QTRAP-MS, and IC. The authors observed various products they provided chemical formula and compared against ambient and lab studies where these compounds had been observed before. Further, they determined the heterogenous OH oxidation of apOS-249 and found the lifetime to be comparable to the lifetime of aerosol (~10 days). They explored potential reaction mechanisms. Finally, they investigated the potential amount of inorganic sulfate (SO4) produced from the heterogenous reaction of apOS-249 with OH. The paper is generally well written with interesting results that would be of importance for the community that reads ACP. I recommend publication after the authors address some of the comments below.*

**We thank the reviewer for his/her thoughtful comments. The referee's comments are below in italics followed by our responses in normal font.**

**Comment #1:**
*1. In general, more information concerning the oxidation flow reactor (OFR) is needed to better understand and replicate the experiments. The following details needs to be included:*
*1a) How was the aerosol and gases introduced into the system? How were they sampled from the OFR?*
*1b) What material is the OFR made of?*
*1c) Was this made in laboratory or purchased from a company?*
*1d) What was the temperature for the experiments?*
*1e) How was OH reactivity determined during the experiments?*
*1f) What were the losses within the OFR? Was this included in the calculations?*

**Author Response:**
Thanks for the suggestions. We have revised the experimental method in the manuscript and provided more details in the Supporting Information (**Sect. S1 Experimental Details** in SI). A simplified schematic diagram of our experimental setup was also added in the Supporting Information as **Scheme S1**. Please see our responses to specific questions below, followed by corresponding changes made in manuscript and SI (highlighted in blue).

*1a) How was the aerosol and gases introduced into the system? How were they sampled from the OFR?*
**Response:** We have prepared a simple schematic diagram of our experimental setup. αpOS-249 aerosols were first generated by passing its solution through a constant output atomizer (TSI Model 3076) using 3 L min$^{-1}$ of nitrogen (N$_2$). Before entering the reactor, the aerosol stream was mixed with nitrogen (N$_2$), oxygen (O$_2$), and ozone (O$_3$) to make up a total flow of ~5 L min$^{-1}$, corresponding to a residence time of ~156 s. Aerosols leaving the reactor were collected into Teflon filters (2.0 μm pore size, Pall Corporation) at a sampling flow rate of 3 L min$^{-1}$ conducted by an air sampling pump (Gilian 500, Sensidyne) for 30 min.

*1b) What material is the OFR made of?*
**Response:** The OFR was made of aluminum with a volume of approximately 13 L (18-inch length and 8-inch inner diameter) (Kang et al., 2007).

*1c) Was this made in laboratory or purchased from a company?*
**Response:** The OFR (also called Potential Aerosol Mass, PAM) was designed and obtained from Prof. William Brune's group at Penn State.

*1d) What was the temperature for the experiments?*
**Response:** The experiment was conduct at $298.0 \pm 0.5$ K.

*1e) How was OH reactivity determined during the experiments?*
**Response:** As mentioned in the manuscript, OH was generated via photolysis of $O_3$ in the presence of water vapor. $O_3$ was generated by passing $O_2$ through an $O_3$ generator (ENALY 1000BT-12). The OH reactivity was determined by measuring the decay of sulfur dioxide ($SO_2$) in independent calibrating experiments (Teledyne $SO_2$ analyzer, Model T100) based on the reaction rate constant between gas-phase OH radicals and $SO_2$ (= $9.0 \times 10^{-13}$ molecule$^{-1}$ cm$^3$ s$^{-1}$) at 298 K (Kang et al., 2007) and was represented by the OH exposure, which is equivalent to the product of gas-phase OH concentration and the residence time. The OH exposure ranged from 0 to $17.4 \times 10^{11}$ molecules cm$^{-3}$ s in this study.

*1f) What were the losses within the OFR? Was this included in the calculations?*
**Response:** The OFR was designed with a small surface-to-volume ratio to minimize the aerosol wall loss (Kang et al., 2007; Lambe et al., 2011). The aerosol transmission efficiency for aerosol diameter larger than 150 nm was reported to be greater than 80% (Lambe et al., 2011). In our study, the wall loss was expected to be small as the aerosol diameter was measured to be 181.3 nm and the aerosols with a diameter larger than 150 nm accounted for a significant fraction of total aerosol number and mass. This wall loss factor was not corrected in the calculations. However, we expect this would not affect the determination of reaction kinetics (i.e. $k$) and inorganic sulfate yield. This is because concentration ratios (e.g. $I/I_0$) were used and the effect of wall loss would be cancelled out in the calculations.

References:
- Kang, E., Root, M. J., Toohey, D. W., and Brune, W. H.: Introducing the concept of Potential Aerosol Mass (PAM), Atmos. Chem. Phys, 7, 5727–5744, https://doi.org/10.5194/acp-7-5727-2007, 2007.
- Lambe, A. T.; Ahern, A. T.; Williams, L. R.; Slowik, J. G.; Wong, J. P. S.; Abbatt, J. P. D.; Brune, W. H.; Ng, N. L.; Wright, J. P.; Croasdale, D. R.; Worsnop, D. R.; Davidovits, P.; Onasch, T. B.: Characterization of Aerosol Photooxidation Flow Reactors: Heterogeneous Oxidation, Secondary Organic Aerosol Formation and Cloud Condensation Nuclei Activity Measurements, Atmos. Meas. Tech, 4, 445−461, https://doi.org/10.5194/amt-4-445-2011, 2011.

**Revision in manuscript:**
Page 5, Line 7: "Heterogeneous OH oxidation of αpOS-249 aerosols was carried out using a 13-L aluminium OFR at $50 \pm 2.0$ % RH and $298.0 \pm 0.5$ K. Experimental details together with schematic diagram (**Scheme S1**) are given in the Supporting Information. Briefly, αpOS-249 was first dissolved in deionized water (0.1 wt %) followed by a 30-min sonication. Aqueous aerosols were generated by passing the solution through an atomizer (TSI Model 3076) using 3 L min$^{-1}$ of nitrogen ($N_2$). The aerosol stream was then directly mixed with ozone ($O_3$), wet/humidified nitrogen ($N_2$) and oxygen ($O_2$) to control the RH. A total flow of ~5 L min$^{-1}$ was fed into the reactor, corresponding to a residence time of ~156 s (Xu et al., 2020a)."

Page 5, Line 19: "The OH exposure, a product of gas-phase OH radical concentration and the residence time, was in the range of 0–$17.4 \times 10^{11}$ molecule cm$^{-3}$ s. It was determined by measuring the decay of sulfur dioxide ($SO_2$) in independent calibrating experiments (Teledyne $SO_2$ analyzer, Model T100) based on the reaction rate between gas-phase OH radicals and $SO_2$ (= $9.0 \times 10^{-13}$ molecule$^{-1}$ cm$^3$ s$^{-1}$) at 298 K (Kang et al., 2007)."

**Revision in SI:**
Page 2, Line 1: "
**1.  Experimental Details**

The heterogeneous OH oxidation of αpOS-249 aerosols was conducted using an OFR with a volume of ~ 13 L (18-inch length, 8-inch inner diameter) at $50 \pm 2.0$ % RH and $298.0 \pm 0.5$ K. As shown in Scheme S1, aqueous αpOS-249 aerosols were first generated by passing its solution through a constant output atomizer (TSI Model 3076) using 3 L min$^{-1}$ of nitrogen ($N_2$). Before entering the reactor, the aerosols were directly mixed with dry/wet nitrogen ($N_2$), oxygen ($O_2$) and ozone ($O_3$) to make up a total flow of ~ 5 L min$^{-1}$, corresponding to a residence time of ~ 156 s. The relative humidity (RH) inside the reactor was maintained by varying the mixing ratio of dry and humidified $N_2$. The RH and temperature were measured by a RH-temperature sensor (Vaisala, HM40).

10        Inside the reactor, OH was generated via photolysis of $O_3$ with UV light at 254 nm in the presence of water vapor. The $O_3$ was generated by passing $O_2$ through an $O_3$ generator (ENALY 1000BT-12). The concentration of gas-phase OH radical was varied by changing the $O_3$ concentrations, monitored by an $O_3$ analyzer (2B technologies, Model 202). The OH exposure, a product of gas-phase OH radical concentration and the residence time, ranged from $0-17.4 \times 10^{11}$ molecule cm$^{-3}$ s and was determined by measuring the decay of sulfur dioxide ($SO_2$) (Teledyne $SO_2$ analyzer, Model T100) in independent calibrating experiments in the absence of αpOS-249 aerosols based on the reaction rate between gas-phase OH radicals and $SO_2$ ($= 9.0 \times 10^{-13}$ molecule$^{-1}$ cm$^3$ s$^{-1}$) at 298 K (Kang et al., 2007). Furthermore, $SO_2$ calibration experiments in the presence of αpOS-249 aerosol were also conducted to investigate the effects of the aerosols on the generation and concentration of gas-phase OH radicals inside the

20 reactor. A variation of ~10 % in the determination of OH exposure was observed over the experimental conditions.

       The aerosol stream leaving the reactor then passed through an annular Carulite catalyst denuder (manganese dioxide/copper oxide catalyst; Carus Corp.) and an activated charcoal denuder to remove residual $O_3$ and other gas-phase species. 3 L min$^{-1}$ of the stream was sampled onto the Teflon filters (2.0 μm pore size, Pall Corporation) by an air sampling pump (Gilian 500, Sensidyne) for 30 min, with a total gas sampling volume of ~ 90 L. Duplicate filters were collected from each of oxidation experiments for subsequent chemical analysis. After collection, filters were immediately stored at −20°C in the dark and analysed within 3 months.

30

       Part of the remaining stream was introduced into a scanning mobility particle sizer (SMPS, TSI, CPC Model 3775, Classifier Model 3081) to measure the size distribution of aerosols. The size distribution was sampled from 16 to 604.3 nm and scans were repeated every 180 s (sampling flow of ~ 0.3 L min$^{-1}$ and sheath flow of 3 L min$^{-1}$). The aerosol mass loading was determined from measured volume concentration assumed for spherical aerosols with a unit density since the density of αpOS-249 is not available. As the sodium salts of the organosulfates ($R-OSO_3Na$) usually have density larger than 1.0 g cm$^{-3}$ (e.g. 1.60 g cm$^{-3}$ of $CH_3SO_4Na$; 1.46 g cm$^{-3}$ of $C_2H_5SO_4Na$; Chemistry Dashboard), the reported aerosol mass loadings are considered as low limits. Before oxidation, the mean surface weighted diameter for aerosol distribution was about $181.3 \pm 0.5$ nm with a geometric standard

40 deviation of 1.3 and the aerosol mass loading was measured to be ~2000 μg m$^{-3}$.

       The OFR was designed with a small surface-to-volume ratio to minimize the aerosol wall loss (Kang et al., 2007; Lambe et al., 2011). The aerosol transmission efficiency for aerosol diameter larger than 150 nm was reported to be greater than 80% (Lambe et al., 2011). In our study, the wall loss was expected to be small as the aerosol diameter was measured to be $181.3 \pm 0.5$ nm and the aerosols with a diameter larger than 150 nm accounted for a significant fraction of total aerosol number and mass. This wall loss factor was thus not corrected in the calculations. However, we expect this would not significantly affect the determination of reaction kinetics (i.e. $k$) and inorganic sulfate yield. This is because concentration ratios (e.g. $I/I_0$) were used and the effect of wall loss would be cancelled out in

50 the calculations.

[Figure]

**Scheme S1. Schematic diagram for experimental setup of the heterogeneous OH oxidation.**"

**Comment #2:**
*2. Other information within the methods that would improve the paper:*
*2a) Was a drier used after atomization?*
*2b) How were O3 and gas-phase species removed prior to sampling? What was the particle loss through this method to remove gas-phase species?*
*2c) How was the SMPS operated? What SMPS system was used?*
10 *2d) What assumption was used to assume an aerosol mass loading of 2000 ug m-3? Was this mass concentration too high for heterogenous oxidation (e.g., OH-limited at the beginning due to too high aerosol mass loading compared to OH concentration)?*
*2e) Description of the aerosol collection/filters are needed -- size, type, and pore size of filter, were they cleaned before use, how backgrounds were collected, stability of products on filters, and any impactors that may have been used.*
*2f) A simple diagram or an actual photo of the experimental set-up would be beneficial.*

**Author Response:**
We agree with the reviewer's suggestions. We have prepared point-to-point responses below.
20 Changes have also been made in the manuscript and **Experiment Details** section of the SI provided above.

2a) *Was a drier used after atomization?*
**Response:** In our experiments, aqueous droplets generated by the atomizer did not pass through a diffusion dryer and were directly mixed with gases such as humidified nitrogen, oxygen, ozone before entering the flow tube reactor.

2b) *How were O3 and gas-phase species removed prior to sampling? What was the particle loss through this method to remove gas-phase species?*
30 **Response:** An annular Carulite catalyst denuder and an activated charcoal denuder was used to remove $O_3$ and gas-phase species from the aerosol stream, respectively. We have measured the aerosol loss before and after the denuders under 50 % RH. Only 8.50 % and 2.04 % of differences were observed in aerosol number and mass concentration, respectively. We also would like to note that the size measurements of the aerosols were carried out after the denuders (**Scheme S1**).

2c) *How was the SMPS operated? What SMPS system was used?*
**Response:** A scanning mobility particle sizer (SMPS, TSI, CPC Model 3775, Classifier Model 3081) was used to measure the aerosol size and number distribution. The size distribution was sampled from 16 to 604.3 nm and scans were repeated every 180 s (sampling flow of ~ 0.3 L $min^{-1}$ and sheath flow
40 of 3 L $min^{-1}$).

2d) *What assumption was used to assume an aerosol mass loading of 2000 ug m-3? Was this mass concentration too high for heterogenous oxidation (e.g., OH-limited at the beginning due to too high aerosol mass loading compared to OH concentration)?*
**Response:** Thanks for the comment. The aerosol mass loading was determined from measured volume concentration assumed for spherical aerosols with a unit density. We don't correct for the aerosol density since the density of αpOS-249 is not known. From the literatures, the sodium salts of the organosulfates (R−OSO$_3$Na) usually have the density larger than 1.0 g $cm^{-3}$ (e.g. 1.60 g $cm^{-3}$ of

CH$_3$SO$_4$Na; 1.46 g cm$^{-3}$ of C$_2$H$_5$SO$_4$Na; Chemistry Dashboard). We would like to consider the reported aerosol mass loadings as low limits.

In our experiment, the gas-phase OH radicals were generated by the photolysis of O$_3$ under UV light ($\lambda$ = 254 nm) illumination in the presence of water vapor. O$_3$ was generated by passing the O$_2$ through an ozone generator. The OH concentration was regulated by changing the O$_3$ concentration and was determined by measuring the decay of sulfur dioxide (SO$_2$) in independent experiments in the presence or absence of aerosols. In the absence of aerosols, the measured OH concertation ranged from 3.06 × 10$^9$ molecules cm$^{-3}$ to 1.11 × 10$^{10}$ molecules cm$^{-3}$. We also found that the presence of aerosols did not significantly affect the generation of gas-phase OH radicals and the determination of OH exposure, with differences less than ∼10 % at each oxidation level. This may suggest that the oxidation would not be limited by the OH concentration.

*2e) Description of the aerosol collection/filters are needed -- size, type, and pore size of filter, were they cleaned before use, how backgrounds were collected, stability of products on filters, and any impactors that may have been used.*
**Response:** Thanks for the comment. In our study, aerosols leaving the reactor were collected into Teflon filters (47mm, 2.0 μm pore size, Pall Corporation) through filtration at a sampling flow rate of 3 L min$^{-1}$ using an air sampling pump (Gilian 500, Sensidyne) for 30 min. These filters were used as purchased while the filter holder was cleaned prior to collection.

For the background, before the experiments, the OFR was flushed using high purity gases (N$_2$ and O$_2$) and then exposed to a high concentration of OH for several hours until background aerosol mass loading was less than a few μg m$^{-3}$. Given the low concentration of aerosols in the background, the background signals were considered to be insignificant.

For the stability of products, after collection, these filters were immediately stored at −20℃ in the dark and analyzed within 3 months. The extraction efficiency of αpOS-249 was determined to be ∼85 %. This high recovery suggests that αpOS-249 is relatively stable in our sample preparation and extraction procedure. We did not know the recovery of the reaction products as their standards were not available. a recent study by Hughes et al. (2019) evaluated the stability of a range of OSs (e.g. methyl sulfate, hydroxyacetone sulfate, two α-pinene derived OSs: $m/z$ = 279 (C$_{10}$H$_{15}$O$_7$S$^-$), and $m/z$ = 281 (C$_{10}$H$_{17}$O$_7$S$^-$)) on filters frozen at −20℃ over the course of one year and extracted via similar procedure and analyzing using HPLC-ESI-HRMS. They found that stored OSs samples showed no degradation during the one year of storage and thus suggested that OSs with different alkyl, carboxylate and hydroxyl functional groups likely remain stable after collection onto filters when stored frozen. In addition, Wang et al. (2017) reported that there was no degradation over two-years' storage for αpOS-249 as well as limonene OS ($m/z$ = 249 (C$_{10}$H$_{17}$O$_5$S$^-$)) and limonaketone OS ($m/z$ = 249 (C$_9$H$_{15}$O$_6$S$^-$)), which have similar carbon skeletons while possessing extra functional (ketone or double bond) groups. Altogether, we would suggest that reaction products are likely stable after collection onto filters during the storage at −20℃. However, further study is warrant for the investigation of stability of the reaction products.

The following changes were made in the manuscript, and details can be referred to the SI as replied to Comment #1.
Page 5, Line 19: "The OH exposure, a product of gas-phase OH radical concentration and the residence time, was in the range of 0–17.4 × 10$^{11}$ molecule cm$^{-3}$ s. It was determined by measuring the decay of sulfur dioxide (SO$_2$) in independent calibrating experiments (Teledyne SO$_2$ analyzer, Model T100) based on the reaction rate between gas-phase OH radicals and SO$_2$ (= 9.0 × 10$^{-13}$ molecule$^{-1}$ cm$^3$ s$^{-1}$) at 298 K (Kang et al., 2007). It acknowledges that the presence of aerosols did not significantly affect the generation of gas-phase OH radicals and the determination of OH exposure (less than ∼10 %). The aerosol stream leaving the reactor passed through an annular Carulite catalyst denuder (manganese dioxide/copper oxide catalyst; Carus Corp.) and an activated charcoal denuder to remove residual O$_3$ and other gas-phase species. Aerosols were collected onto the Teflon filters (47mm, 2.0 μm pore size, Pall Corporation) through filtration at a sampling flow rate of 3 L min$^{-1}$ using an air sampling pump

(Gilian 500, Sensidyne) for 30 min, with a total gas sampling volume of ~ 90 L. Duplicate filters were collected from each of oxidation experiments for subsequent chemical analysis. After collection, filters were immediately stored at −20 °C in the dark and analysed within 3 months. Part of the remaining stream was introduced into a scanning mobility particle sizer (SMPS, TSI, CPC Model 3775, Classifier Model 3081) to measure the size distribution of the aerosols. The aerosol mass was determined from measured volume concentration assumed for spherical aerosols with a unit density."

Page 8, Line 10: "The high recovery of αpOS-249 suggests the sample preparation and extraction methods are effective. Wang et al. (2017) have also reported that there was no degradation for αpOS-249 after two-years' storage at low temperature (−20°C). A recent study by Hughes et al. (2019) examined the stability of a range of OSs (e.g. methyl sulfate, hydroxyacetone sulfate, two α-pinene derived OSs: $m/z$ = 279 ($C_{10}H_{15}O_7S^-$), and $m/z$ = 281 ($C_{10}H_{17}O_7S^-$)) on filters frozen at −20°C over one year. The filters were extracted via similar procedure applied in this study and the extracts were analyzed by HPLC-ESI-HRMS. They found that the investigated OSs with different functional groups (e.g. alkyl, carboxylate, and hydroxyl groups) showed no degradation during the storage. Taken together, αpOS-249 and its oxidation products (i.e. OSs) which have similar carbon skeletons while possessing different functional groups (alcohol and/or ketone) are likely stable during the storage and pre-treatment processes for chemical analysis."

2f) *A simple diagram or an actual photo of the experimental set-up would be beneficial.*
**Response:** Thanks for the suggestion. We have prepared a simple schematic diagram of our experimental setup and have added this figure to the supporting information (**Scheme S1**). Please see our response above to Comment #1.

**Comment #3:**
*3) As reviewer #1 mentioned, sample preparation and/or sampling of the products with ESI may lead to side reactions. Though many of the products may be hard to synthesize, have surrogates been used to investigate their stability on filter, during preparation, and during sampling? One thing that would help with this question is stating that apOS-249 shows high extraction efficiency (pg 8, ln4-5) sooner and maybe discuss other organosulfates.*

**Author Response:**
Thanks for this insightful comment. We agree with reviewer's comment that sample preparation and/or sampling of the αpOS-249 and reaction products with ESI may bring potential issues for identification and quantification of OSs. With regards to the side reactions (i.e. reactions between organics and solvent) during preparation and electrospray processes, as replied to the concern of reviewer #1, these reactions are considered to be negligible based on the insignificant peak intensities of potential reaction products in our aerosol mass spectra. Detailed discussions have been given in our responses to the reviewer #1 (Comment #1).

To our best knowledge, standards are not yet available for investigating the stability on filter, during preparation, and during sampling of the reaction products. However, we have discussed the stability of the products using their surrogates to our best knowledge (Please see our responses to the reviewer's Comment #2e). For products' stability on filters after collection, a recent study by Hughes et al. (2019) evaluated the stability of a range of OSs (e.g. methyl sulfate, hydroxyacetone sulfate, two α-pinene derived OSs: $m/z$ = 279 ($C_{10}H_{15}O_7S^-$), and $m/z$ = 281 ($C_{10}H_{17}O_7S^-$)) on filters frozen at −20°C over the course of one year and extracted via similar procedure and analyzing using HPLC-ESI-HRMS. They found that stored OSs samples showed no degradation during the one year of storage and thus suggested that OSs with different alkyl, carboxylate and hydroxyl functional groups remain stable after collection onto filters when stored frozen. According to the synthesis reference for αpOS-249 by Wang et al. (2017), there is no degradation during over two-years' storage for standard αpOS-249 as well as limonene OS ($m/z$ = 249 ($C_{10}H_{17}O_5S^-$)) and limonaketone OS ($m/z$ = 249 ($C_9H_{15}O_6S^-$)), which have similar carbon skeletons while possessing extra functional (ketone or double bond) groups. Altogether, reaction products are expected to be stable after collection onto filters during the storage at −20°C.

We agree with the reviewer's suggestion and discussed this issue in earlier section of the manuscript. The following information is added in the manuscript.

Page 8, Line 10: "The high recovery of αpOS-249 suggests the sample preparation and extraction methods are effective. Wang et al. (2017) have also reported that there was no degradation for αpOS-249 after two-years' storage at low temperature (−20℃). A recent study by Hughes et al. (2019) examined the stability of a range of OSs (e.g. methyl sulfate, hydroxyacetone sulfate, two α-pinene derived OSs: $m/z = 279$ ($C_{10}H_{15}O_7S^-$), and $m/z = 281$ ($C_{10}H_{17}O_7S^-$)) on filters frozen at −20℃ over one year. The filters were extracted via similar procedure applied in this study and the extracts were analyzed by HPLC-ESI-HRMS. They found that the investigated OSs with different functional groups (e.g. alkyl, carboxylate, and hydroxyl groups) showed no degradation during the storage. Taken together, αpOS-249 and its oxidation products (i.e. OSs) which have similar carbon skeletons while possessing different functional groups (alcohol and/or ketone) are likely stable during the storage and pre-treatment processes for chemical analysis.

We note that organic compounds such as carbonyls and carboxylic acids could undergo reactions with methanol during extraction, storage, and possibly during the electrospray process (Bateman et al., 2008). For instance, Batman et al (2008) suggested carboxylic acids could react with methanol to form esters and with carbonyls to hemiacetals and acetals. We checked the presence and relative abundance of these potential products (to that of our precursor, αpOS-249) in our aerosol mass spectra. At the maximum OH exposure, only a few products that could be potentially formed from the reactions of αpOS-249 with methanol were detected and they had negligible intensities. This would suggest that the influence of methanol is not significant on the identification of the major reaction products."

**Comment #4:**
*4) Section 3.1. These results are really interesting and important for the community. However, as has been studied more within the community, phase state, ionic strength, and "shell" formation are most likely extremely important parameters in the OH heterogenous reaction of apOS-249. As this is a single phase experiment, there is likely little phase state and shell formation concerns (which might be useful to explore in future experiments including with changing RH). I strongly recommend the authors briefly put this experiment into this context and how these results are most likely upper limits.*
*Similar thoughts in regards to ionic strength, especially as this may impact the reaction mechanism and what intermediate and final products are stable.*

**Author Response:**
Thanks for the valuable comments and suggestions. We agree with the reviewer that the complex interplay between aerosol phase, morphology and ionic strength can significantly alter the heterogenous OH reaction kinetics and mechanisms and are warrant for further study. We have added the following discussion in **Sect. 3.1 Oxidation Kinetics** to address these issues.

[revised manuscript text omitted]

- Marshall, F. H., Miles, R. E. H., Song, Y.-C., Ohm, P. B., Power, R. M., Reid, J. P. and Dutcher, C. S.: Diffusion and reactivity in ultra-viscous aerosol and the correlation with particle viscosity, Chem. Sci., 7, 1298–1308, doi:10.1039/c5sc03223g, 2016.

- Marshall, F. H., Berkemeier, T., Shiraiwa, M., Nandy, L., Ohm, P. B., Dutcher, C. S. and Reid, J. P.: Influence of particle viscosity on mass transfer and heterogeneous ozonolysis kinetics in aqueous-sucrose-maleic acid aerosol, Phys. Chem. Chem. Phys., 20, 15560–15573, doi:10.1039/c8cp01666f, 2018.

- McNeill, V. F., Wolfe, G. M. and Thornton, J. A.: The Oxidation of oleate in submicron aqueous salt aerosols: evidence of a surface process, J. Phys. Chem. A, 111, 1073–1083, doi:10.1021/jp066233f, 2007.

- McNeill, V. F., Yatavelli, R. L. N., Thornton, J. A., Stipe, C. B. and Landgrebe, O.: Heterogeneous OH oxidation of palmitic acid in single component and internally mixed aerosol particles: vaporization and the role of particle phase, Atmos. Chem. Phys., 8, 5465–5476, doi:10.5194/acp-8-5465-2008, 2008.

- Mungall, E. L., Wong, J. P. S., and Abbatt, J. P. D.: Heterogeneous Oxidation of Particulate Methanesulfonic Acid by the Hydroxyl Radical: Kinetics and Atmospheric Implications, ACS Earth Space Chem., 2, 48−55, 10.1021/acsearthspacechem.7b00114, 2017.

- Qiu, Y. and Molinero, V.: Morphology of liquid-liquid phase separated aerosols, J. Am. Chem. Soc., 137, 10642–10651, doi:1021/jacs.5b05579, 2015.

- Shiraiwa, M., Ammann, M., Koop, T., and Pöschl, U.: Gas uptake and chemical aging of semisolid organic aerosol particles, Proc. Natl. Acad. Sci., 108, 11003–11008, https://doi.org/10.1073/pnas.1103045108, 2011.

- Shiraiwa, M., Zuend, A., Bertram, A. K., and Seinfeld, J. H.: Gas–particle partitioning of atmospheric aerosols: Interplay of physical state, non-ideal mixing and morphology, Phys. Chem. Chem. Phys., 15, 11441-11453, https://doi.org/10.1039/C3CP51595H, 2013.

- Slade, J. H. and Knopf, D. A.: Multiphase OH oxidation kinetics of organic aerosol: The role of particle phase state and relative humidity, Geophys. Res. Lett., 41, 5297–5306, doi:10.1002/2014gl060582, 2014.

- Xu, R., Lam, H. K., Wilson, K. R., Davies, J. F., Song, M., Li, W., Tse, Y.-L. S., and Chan, M. N.: Effect of inorganic-to-organic mass ratio on the heterogeneous OH reaction rates of erythritol: implications for atmospheric chemical stability of 2-methyltetrols, Atmos. Chem. Phys., 20, 3879–3893, https://doi.org/10.5194/acp-20-3879-2020, 2020.

- You, Y., Smith, M. L., Song, M., Martin, S. T. and Bertram, A. K.: Liquid–liquid phase separation in atmospherically relevant particles consisting of organic species and inorganic salts, Int. Rev. Phys. Chem., 33, 43–77, doi:10.1080/0144235x.2014.890786, 2014.

**Comment #5:**

*5) Section 3.4. Though the authors show that sulfate formation is minimally important, there is a question if the sulfate they observe may be due to the hydrolysis of the primary, secondary, etc. products. The authors investigate and determine that apOS-249 is likely stable but it is currently not clear if the other products may be.*

**Author Response:**

Thanks for this insightful comment. We agree with the reviewer that inorganic sulfate could also be generated from the hydrolysis of reaction products. This is because secondary and tertiary OS could readily undergo hydrolysis and may form upon OH oxidation based on our proposed reaction mechanisms. We have added this information in the manuscript.

Page 20, Line 22: "Furthermore, we cannot rule out the possibility of the formation of $SO_4^{2-}$ could be from the hydrolysis of reaction products as some secondary and tertiary OS could readily undergo hydrolysis and may form upon oxidation."